# A systematic review of the effect of The Daily Mile™ on children's physical activity, physical health, mental health, wellbeing, academic performance and cognitive function

**Gavin Breslin**[1,2], **Medbh Hillyard**[1,2]*, **Noel Brick**[2⊙], **Stephen Shannon**[1,2,3⊙], **Brenda McKay-Redmond**[4‡], **Barbara McConnell**[4‡]

**1** Bamford Centre for Mental Health and Wellbeing, School of Psychology, Ulster University, Coleraine, Northern Ireland, **2** School of Psychology, Ulster University, Coleraine, Northern Ireland, **3** Sport and Exercise Sciences Research Institute, Ulster University, Belfast, Northern Ireland, **4** Early Childhood Studies Department, Stranmillis University College, Belfast, Northern Ireland

⊙ These authors contributed equally to this work.
‡ BMR and BM also contributed equally to this work.
* m.hillyard@ulster.ac.uk

## Abstract

### Background

A minority of children in the United Kingdom meet the recommended physical activity guidelines. One initiative which has been introduced to try and increase the physical activity levels of school children is The Daily Mile™ (TDM). The aim of this review was to determine the effect of TDM on children's physical activity levels, physical health, mental health, wellbeing, academic performance and cognitive function.

### Methods

Six databases were systematically searched from TDM's inception (2012) to 30th June 2022. Studies were included if they involved school-aged children (aged 4–12 years), taking part in TDM and measured at least one pre-defined outcome.

### Results

Thirteen studies were included from the 123 studies retrieved. Longer-term participation in TDM was found to increase moderate-to-vigorous physical activity and physical fitness. None of the studies reported a significant change in Body Mass Index or academic performance. An acute bout of TDM was not found to improve cognitive function, however one good-quality study reported that longer-term participation in TDM increased visual spatial working memory. There was evidence from one fair-quality design study that TDM can improve mental health in the short term. There were no significant effects on wellbeing, however scores on self-perceptions improved mainly for children with low baseline self-perceptions.

**Data Availability Statement:** All relevant data are within the paper and its Supporting Information files.

**Funding:** This project was funded by The Education Authority Northern Ireland. The funder had no role in study design, data collection and analysis, decision to publish, or preparation of the manuscript.

**Competing interests:** I have read the journal's policy and the authors of this manuscript have the following competing interests: All authors are members of The Daily Mile Network Northern Ireland. We can confirm that our membership of the Daily Mile Network Northern Ireland does not alter our adherence to PLOS ONE policies on sharing data and materials.

## Conclusion

There is evidence to show that TDM can increase physical activity and physical fitness. However, higher-quality research, with adequate participant randomisation and longer-term, post-intervention follow-up is needed to ensure that any changes accurately reflect the components of TDM and are sustained beyond an intervention time frame. Policy recommendations of TDM increasing PA levels in the short term are supported by the evidence in this review. However, long-term improvement on mental health, wellbeing, academic performance and cognitive function requires further good-to excellent quality research. Promisingly, several protocol articles that include randomised controlled trials with long term follow-up have been published. These higher-quality design studies may provide a stronger evidence-base on the effects of TDM on children's health and should underpin future recommendations in public health policy.

## Systematic review registration

PROSPERO CRD42022340303.

## Background

There is convincing scientific evidence to support the benefits of promoting regular physical activity (PA) to enhance children's health. Health benefits from PA participation include improving children's fitness [1], maintaining healthy weight [2], strengthening muscles and bones, improving sleep quality, and mental health and wellbeing enhancements [3]. There is also some evidence that children who are active have improved cognitive outcomes, such as academic performance and executive function [4]. In addition, children who are substantially active during childhood and adolescence are more likely to maintain their physical activity behaviour through adulthood [5]. Therefore, evidence highlights the importance of providing children and young people with suitable opportunities for PA.

Recommendations from the United Kingdom (UK) Chief Medical Officers (CMO) is that children and young people (5–18 years) should engage in at least 60-minutes per day of Moderate-to-Vigorous Physical Activity (MVPA) [6]. However, despite the known benefits of PA, between 20–44.6% of children aged 5–16 years are currently meeting the recommended level of PA in the UK [7], with children from socially disadvantaged backgrounds being less likely to meet the PA recommendations [8, 9]. For example in Northern Ireland where only 20% of children from a low socio-economic status meet the recommended 60 minutes of MVPA per day [10]. The World Health Organisation's (WHO) Global Action Plan on Physical Activity 2018–2030 aims to reduce the global prevalence of physical inactivity by 10% by 2025 and by a further 5% by 2030 [11].

PA levels are found to increase in children between the ages of three and six [12]. This is due to developmental changes leading to improvements in motor skills and co-ordination along with growing language skills providing greater opportunities for interactive play with caregivers and peers [13]. However, levels then begin to decrease from age six. A recent study found that on average, levels of MVPA decrease by 2.2 minutes/day/year between the ages of six and eleven (95% CI 1.9 to 2.5) [14]. Furthermore, children aged 15 years are less likely to meet the PA guidelines than children aged nine [15]. As most children attend school regularly, the school setting provides a suitable environment to intervene to try and increase the PA levels of children from a wide range of backgrounds [2].

The lack of sufficient time is reported as one of the most prevalent barriers for teachers when attempting to implement PA interventions [16]. One initiative which has been introduced in an attempt to overcome these challenges and increase the PA levels of school-aged children is The Daily Mile (TDM; www.thedailymile.co.uk) [17]. TDM began in Scotland in 2012 and involved children walking, running or wheeling outside for 15 minutes (approximately one mile) on a minimum of three days of the week [17]. TDM is shaped around ten core delivery principles to enable implementation. TDM core principles state that it should take place in addition to Physical Education time and should happen during curriculum time, and therefore not during lunch or break time. It should be quick, taking 15 minutes in total, with no time spent changing or setting up. TDM is intended to be inclusive, and every child should be able to take part and go at their own pace. It should be carried out regardless of the weather on a firm mud-free surface where possible and should be fun for the children. [17]. The practical premise behind TDM is that it is easy to set up, requires no additional equipment, and can be easily integrated into the school day [17]. Demonstrating its popularity, schools from across 87 countries have signed up to TDM, with over 10,000 schools across the UK signed up [17]. Currently in Northern Ireland 490 schools (over 50%) have signed up on the Daily Mile Foundation website.

The UK Childhood Obesity Plan encourages every school to implement an active mile initiative such as The Daily Mile [18]. However, despite the large number of participating schools, and Government recommendations, the scientific evidence to support the effectiveness of TDM is mixed and arguably limited due to the short-term follow-up in available research studies, and in some cases limited research designs [19, 20]. To justify the inclusion of TDM into policy, there is a need for a strong evidence-base to support the health benefits of TDM. This evidence-base is currently lacking, however [21]. Furthermore, with numerous schools incorporating TDM into their COVID-19 recovery plans with the aim to improve children's mental health and wellbeing, it is important to understand what effects, if any, the TDM can have beyond increasing PA. Failing to provide such evidence exacerbates the potential risk of a futile policy attempt to increase children's activity. As such, it is surprising that no systematic review of TDM has been conducted to date to inform policy decision making.

To respond to the lack of any review of TDM, this systematic review will identify the published literature on TDM, evaluate their methodological quality, and summarise the findings of the available evidence for TDM. Specifically, the review will determine the effect of TDM on 4–12 year old school children's PA levels, physical health, mental health, wellbeing, academic performance and cognitive function. The practical application of TDM in schools, the implications for policy makers and directions for further research are discussed.

## Methods

### Review question

This review aimed to answer the following question:

1. What are the effects of participating in The Daily Mile on children's physical activity levels, physical health, mental health, wellbeing, academic performance and cognitive function?

### Protocol

The review was registered on the International prospective register of systematic reviews (PROSPERO Registration Number: CRD42022340303). The Preferred Reporting Items for Systematic Reviews and Meta-Analyses (PRISMA) guidelines [22] were followed. A PRISMA checklist is included as S1 Checklist.

## Eligibility criteria

**Types of studies.** For inclusion in the review, studies were published in a peer-reviewed academic journal and written in English language. Conference abstracts and grey literature were not eligible for inclusion. All study designs were included; Randomised controlled trials, quasi-experimental studies, pilot studies, repeat measures, cross-sectional, and natural experiments. The eligibility criteria were structured around the Population, Intervention, Control, Outcome (PICO) framework.

**Population.** The population was school-aged children, between the ages of 4 and 12. If children attended a special education needs school, they were also eligible for inclusion.

**Intervention.** The intervention had to consist of TDM initiative. If a study included TDM initiative alongside or in conjunction with another intervention, the study was excluded as it was not possible to determine the independent effects of TDM on the specified outcomes. Studies with PA interventions described as similar to TDM, but not specifically TDM were not included. This was to ensure included interventions were based upon the principles of TDM outlined earlier.

**Control.** Studies were included if they contained control or comparison groups, but this was not a requirement. Due to the limited number of studies carried out on TDM, it was deemed important to keep the inclusion criteria for study type broad.

**Outcomes.** For studies to be included, they had to measure at least one outcome pertaining to the following six categories: Physical activity (PA) levels (self-report or objective measures [accelerometers, pedometer worn devices]), physical fitness (e.g. Multistage Fitness Test), indicators of physical health status (e.g. weight, body mass index [BMI], body composition), mental health, psychological wellbeing, academic performance, and cognitive function.

**Definitions.** For this review the following outcomes were defined as follows:

- Cognitive function: Mental processes such as executive function which may impact academic performance [23].

- Academic achievement: A child's performance on tasks performed at school, often described in the form of grades or results of standardised tests [23].

- Wellbeing: A positive state experienced by individuals and societies, similar to health and a resource for daily life. It is determined by social, economic and environmental conditions [24].

## Information sources and search strategy

A systematic search of six electronic databases (MEDLINE, Embase, Web of Science, PsycINFO, SPORTDiscus and Scopus) was conducted. The search timeframe was 2012 (the year of TDM 's inception) to the date of the search (30th June 2022).

Keywords were used in the searches, with truncation and MeSH terms used depending on the database. The search strategy was developed by the authors, alongside the institute librarian. Search terms were divided into three categories; school, TDM and outcomes (see Table 1). The reference lists of eligible studies were also hand searched and Google Scholar was searched using key words for any available studies.

## Study selection

All references retrieved from the electronic databases were imported into Covidence, a web-based systematic review software programme (Covidence systematic review software, available at: https://www.covidence.org). Covidence automatically removes duplicate articles, these

**Table 1. Search terms used in PsycINFO search (adapted for the other databases).**

| Category | Key terms |
|---|---|
| School | Elementary Schools/ or School Based Intervention/ or primary school or school based health promotion |
| The Daily Mile | daily mile |
| Outcomes | Physical Fitness or Physical Activit* or Exercis* or Run* or Walk* or jog* or Cognition or cognitive function or academic achievement or academic performance or educational achievement or executive function or Well Being or wellbeing or well-being or Mental Health or psychological or Physical Health* or Health* or Child Health/ or Public Health or Quality of Life |
| Limiters | English language and Year 2012-current |

were then checked by a reviewer (MH) to ensure they were exact duplicates. After de-duplication at least two independent reviewers screened all titles and abstracts to assess for eligibility. Articles which met the eligibility criteria were sourced and full-text articles were uploaded into Covidence. The full-text articles were screened independently against the inclusion and exclusion criteria by all authors. The screening tool used is included in S1 File. At least two independent reviewers screened each article, with any disagreements being resolved through consensus with a third reviewer.

## Data extraction

The review team developed a data extraction form, and a single reviewer (MH) extracted the data. A second reviewer (GB, NB or SS) checked the data extraction. Any disagreements were discussed with the other members of the research team where necessary. Only data relevant to the study was extracted, these included: study aim, study design, timings (how long TDM was implemented), participant demographics, baseline characteristics, outcomes, results and information for quality assessment.

## Methodological quality assessment

The quality of the included studies was assessed using a modified version of the Downs and Black checklist [25]. The checklist includes 27 items which covers reporting, external validity, internal validity (bias), internal validity (confounding) and power [25]. The Downs and Black checklist can be used to assess the methodological quality of both random and non-randomised studies [25]. Randomised studies can score a maximum of 28 and non-randomised studies can score a maximum of 25. Based on the overall score given to a study, they were classified as excellent (26–28); good (20–25); fair (15–19); and poor (≤ 14). These categories for classification have been previously used and reported elsewhere [26–28]. One included study was a process evaluation and therefore it was not appropriate to use the Downs and Black checklist and consequently the relevant sections of the Mixed Methods Appraisal Tool (MMAT) were used [29]. The quality assessment was carried out by two reviewers independently (MH, NB, BMcC or BMcKR). Scores were compared and any disagreements were resolved through discussions and a third reviewer was consulted if required.

## Results

### Search results

The literature search generated 123 articles. Duplicate articles were then removed (n = 85), leaving 38 titles and abstracts for review (Fig 1). Of these, 19 articles did not meet the inclusion

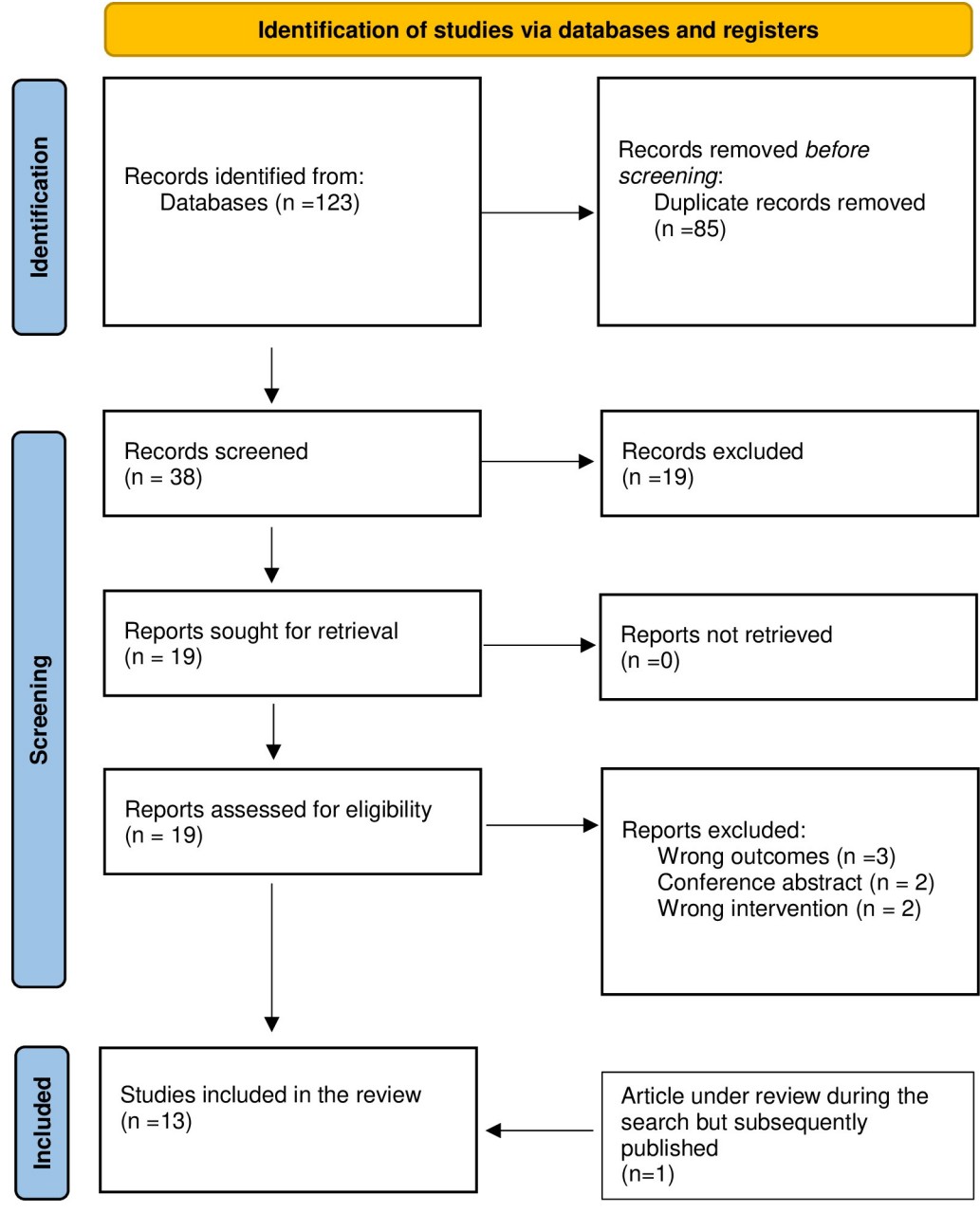

**Fig 1. PRISMA 2020 flow diagram.**

criteria and were excluded. At full text screening another seven articles were excluded. As a result, 12 articles were deemed to meet the inclusion criteria and were included in the review. Studies were excluded because they were the wrong intervention (not TDM) or a combination of interventions, no outcomes of interest were included or were conference abstracts. At the initial search stage, one preprint of a study was identified as meeting all the inclusion criteria, with the exception that it was not peer reviewed, and as a result it was initially excluded. However, it was peer reviewed and published at a later stage (August 2022), therefore was included in the review. A final total of 13 articles were included in the systematic review.

**Table 2. Author, year, study design, duration and sample characteristics of included studies.**

| Author, Year | Study design; duration | Sample characteristics |
|---|---|---|
| Arkesteyn et al. 2022 [30] | Single-arm pilot; 20 weeks | Schools (n = 7), Children (n = 550), 289 males (203 males completed SPPC) age = 9.64 (1.87) |
| Booth et al. 2022 [31] | Quasi experimental between groups; Part of BBC Terrific Scientific project-data collected from schools on participation in TDM and categorised them as no participation, shorter term participation (2 months or less) and longer-term participation (3 months or more). | School classes (n = 503) Children (n = 6908) 50% female, Age = 10.2 (0.7) |
| Breheny et al. 2020 [20] | Cluster RCT; 12 months | Schools (n = 40) Children (n = 2280) (baseline) (47.5% female) Age = 8.9 (1) |
| Brustio et al. 2019 [32] | Quasi experimental Pre/post-test; 3months | 5 schools, Children (n = 795) C = 49.8% female, I = 45.7% female. Age = 8(1) |
| Brustio et al. 2020 [33] | Quasi-experimental; 6 months | N = 548 (49.1% female) Age = 9.14 |
| Chesham et al. 2018 [34] | Repeated measures pilot study; Intervention group-8months, Control 4 months | Schools (n = 2) Children (n = 379) C = 50% female I = 49% female Age = 8.2 (2.0) |
| De Jonge et al. 2020 [35] | Multi-arm, partly RCT with 3 groups; 12 weeks | Schools (n = 9), Children (n = 659) C = 52.1% female, Intervention combined = 50.2% female. Age C = 10.1 (0.1) I = 10.0 (0.1) |
| Dring et al. 2022 [36] | Quasi-experimental: 5 weeks | Schools (n = 2) Children (n = 79) Gender not reported Age = 10.3 (0.8) |
| Harris et al. 2020 [37] | Two-phase multi-method; 3 months | Schools (n = 1) Children (n = 75) Age = 7 years 8months |
| Hatch et al. 2021a [38] | Within subject randomised crossover counterbalanced; single bout of TDM and resting separated by 7 days | Schools (n = 8) Children (n = 104) (46% female) Age = 10.4 (0.7) |
| Hatch et al. 2021b [39] | Cross-sectional descriptive; Single bout of TDM | Schools (n = 8) Children (n = 80) (50% female) Age = 10.4 (0.7) |
| Marchant et al. 2020 [40] | Natural experiment; 3–6 months depending on school | Schools (n = 6), Children (n = 258 imputed) (46% female) Age = 10.2 (0.9) baseline imputed |
| Morris et al. 2019 [19] | RCT; one session of TDM | School classes (n = 14) Children (n = 303) C = 57% female, I = 56% female. Age = 8.99 (0.5) |

SPPC Self-Perception Profile for Children C-control I-Intervention RCT- Randomised Controlled Trial

## Description of studies

The study characteristics are provided in Table 2. Fields included: author's names, year, study design, duration and sample characteristics.

All articles were published between 2018 and 2022. Nine studies were conducted in the UK [19, 20, 31, 34, 36–30], two conducted in Italy [32, 33], one in Belgium [30] and one in The Netherlands [35]. The study sample sizes ranged from 75 participants [37] to 6908 [31]. Nine of the studies included control or comparison groups [19, 20, 31–36, 38]. Four of these nine studies employed some type of randomisation; two of the nine were randomised controlled trials [19, 20], one partial randomisation (schools which had volunteered to implement and perform TDM were randomised into intervention and intervention-plus groups) [35] and one adopted a within subject randomised crossover counterbalanced design [38].

The children in the included studies participated in TDM for different lengths of time, ranging from one session of TDM [19, 38, 39] to 12 months participation [20]. The modal duration of participation was three months [32, 35, 37].

## Methodological quality assessment

The quality of the included studies varied, with scores ranging from 15 [39] to 26 [19]. One study was classified as excellent [19], six as good [20, 31–33, 36, 38], five as fair [30, 34, 35, 39, 40] and

**Table 3. Modified downs and black quality checklist scoring for included studies.**

| Domain | Items | Study | | | | | | | | | | | |
|---|---|---|---|---|---|---|---|---|---|---|---|---|---|
| | | Arkesteyn et al. 2022 [30] | Booth et al. 2022 [31] | Breheny et al. 2020 [20] | Brustio et al. 2019 [32] | Brustio et al. 2020 [33] | Chesham et al. 2018 [34] | de Jonge et al. 2020 [35] | Dring et al. 2022 [36] | Hatch et al. 2021a [38] | Hatch et al. 2021b [39] | Marchant et al. 2020 [40] | Morris et al. 2019 [19] |
| **Reporting** | 1 | 1 | 1 | 1 | 1 | 1 | 1 | 1 | 1 | 1 | 1 | 1 | 1 |
| | 2 | 1 | 1 | 1 | 1 | 1 | 1 | 1 | 1 | 1 | 1 | 1 | 1 |
| | 3 | 1 | 1 | 1 | 1 | 1 | 1 | 1 | 0 | 1 | 1 | 1 | 1 |
| | 4 | 1 | 1 | 1 | 1 | 1 | 1 | 1 | 1 | 1 | 1 | 1 | 1 |
| | 5 | 0 | 2 | 2 | 2 | 2 | 2 | 1 | 0 | 2 | 0 | 2 | 2 |
| | 6 | 1 | 1 | 1 | 1 | 1 | 1 | 1 | 1 | 1 | 1 | 1 | 1 |
| | 7 | 1 | 1 | 1 | 1 | 1 | 1 | 1 | 1 | 1 | 1 | 1 | 1 |
| | 8 | 0 | 0 | 1 | 1 | 0 | 0 | 0 | 0 | 0 | 0 | 0 | 1 |
| | 9 | 1 | 0 | 1 | 0 | 1 | 0 | 0 | 1 | 0 | 0 | 0 | 1 |
| | 10 | 1 | 1 | 1 | 1 | 1 | 1 | 0 | 1 | 1 | 1 | 1 | 1 |
| **External validity** | 11 | 1 | 1 | 1 | 1 | 1 | 0 | 1 | 1 | 1 | 1 | 1 | 1 |
| | 12 | 1 | 1 | 1 | 1 | 1 | 1 | 1 | 1 | 1 | 1 | 1 | 1 |
| | 13 | 1 | 1 | 1 | 1 | 1 | 1 | 1 | 1 | 1 | 1 | 1 | 1 |
| **Internal validity-bias** | 14 | 0 | 0 | 0 | 0 | 0 | 0 | 0 | 0 | 0 | 0 | 0 | 0 |
| | 15 | 0 | 0 | 1 | 0 | 0 | 0 | 0 | 0 | 0 | 0 | 0 | 0 |
| | 16 | 1 | 1 | 1 | 1 | 1 | 1 | 1 | 1 | 1 | 1 | 0 | 1 |
| | 17 | 1 | 1 | 1 | 1 | 1 | 0 | 1 | 1 | 1 | 1 | 0 | 1 |
| | 18 | 1 | 1 | 1 | 1 | 1 | 1 | 1 | 1 | 1 | 1 | 1 | 1 |
| | 19 | 1 | 1 | 0 | 1 | 1 | 0 | 1 | 1 | 1 | 1 | 0 | 1 |
| | 20 | 1 | 1 | 1 | 1 | 1 | 1 | 1 | 1 | 1 | 1 | 1 | 1 |
| **Internal validity-confounding** | 21 | 0 | 1 | 1 | 1 | 1 | 1 | 1 | 1 | 1 | 0 | 1 | 1 |
| | 22 | 1 | 1 | 1 | 1 | 1 | 0 | 1 | 1 | 1 | 0 | 0 | 1 |
| | 23 | 0 | 0 | 1 | 0 | 0 | 0 | 0 | 0 | 1 | 0 | 0 | 1 |
| | 24 | 0 | 0 | 0 | 0 | 0 | 0 | 0 | 0 | 0 | 0 | 0 | 1 |
| | 25 | 0 | 1 | 1 | 0 | 1 | 1 | 0 | 1 | 1 | 0 | 1 | 1 |
| | 26 | 1 | 0 | 1 | 1 | 1 | 0 | 1 | 1 | 1 | 0 | 1 | 1 |
| **Power** | 27 | 0 | 0 | 1 | 0 | 0 | 0 | 1 | 1 | 1 | 0 | 0 | 1 |
| **Total (28)** | | 18 | 20 | 25 | 21 | 22 | 16 | 19 | 20 | 23 | 15 | 17 | 26 |
| **Quality** | | Fair | Good | Good | Good | Good | Fair | Fair | Good | Good | Fair | Fair | Excellent |

none were rated poor. Due to the type of study design, one study was assessed using the Mixed Methods Appraisal Tool [29]. The study assessed using the Mixed Methods Appraisal Tool met 100% of the quality criteria [37]. Modified Downs and Black quality checklist scoring for the included studies are provided in Table 3.

## Delivery mode

There were variations in the implementation and execution of TDM across studies, and not all studies followed the core principles set out by TDM Foundation or were poorly reported (i.e., 15-minutes in length, performed in all weather, at least 3x/week, teacher to decide when to

perform TDM, no change in clothes, aim to jog or run for full 15 minutes). For those studies in line with the core principles, eight studies reported the duration of TDM should be 15-minutes [19, 20, 30, 32–34, 36, 37] and two studies reported 20 minutes duration [38, 39]. Several studies reported allowing teachers to choose the time of day to carry out TDM [20, 30, 32–34]. Additionally, several studies reported the desired frequency of TDM, with three studies encouraging daily participation [20, 30, 36]. One study suggested TDM should be performed on all days without Physical Education lessons [35] and one study stated it should be performed on at least three days of the week [31]. There was also some variation in the specified exercise intensity with some studies reporting children should run or walk [20, 30, 33, 34, 38, 39] and one reported that children were asked to run or jog, only stopping for occasional rests if required [32].

An additional core principle of TDM is that it should be inclusive for every child, and children with mobility difficulties should take part [17]. Two studies reported that children with a physical or intellectual disability were excluded from the study [20, 32], two studies included children with physical or intellectual disabilities in the intervention, but they were excluded from the analysis [19, 33]. One study recruited two special education schools (n = 36), the children took part in the study, but did not complete the Self-Perception Profile For Children (SPPC) [30].

There was a large variation in intervention fidelity, with some studies not measuring compliance and implementation. Those that did, reported compliance in the intervention and intervention plus groups as 88% and 90% respectively [35]. One study reported school level compliance, with two schools performing TDM 3x/week, two schools performing TDM 4x/week and three schools performing it 5x/week [30]. One study reported that all but seven participants out of 79 participated in TDM daily [36].

## Outcomes

There was large variation in the outcome measures reported across the 13 studies. The outcome measures and measurement tools, main findings and any comments of note regarding any methodological issues (i.e., missing data etc) are provided in Table 4.

**Effects on physical activity.** Three studies measured and reported PA levels or PA intensity [19, 34, 37]. One session of TDM resulted in a greater amount of MVPA (10.67±2.74 min) compared to the control group (0.44±0.95 min) and the difference was statistically significant [19]. One study reported PA levels after longer term (8 months) participation in TDM and for MVPA there was a relative increase of 9.1 minutes per day [34]. Harris and colleagues reported that Key Stage 1 children (aged 5 to 7) spent 100% of time during TDM at MVPA and at Key Stage 2 (aged 7 to 11) the children spent 88.1% of TDM at MVPA [37].

**Effects on physical fitness.** Nine studies [20, 31–36, 39, 40] measured and reported physical fitness. A variety of different tests were used and included; 6-minute run test [32, 33], Multi-stage fitness test [34–36, 39, 40], and British Athletics Linear Track Test [20]. Six studies [31–36] reported a significant improvement in fitness in the intervention group compared to the control group. One study categorised children by fitness quartile and reported 'highest fit children ran further than less fit children' (p<0.001) [39]. One study compared the shuttle run distances completed by children categorised as deprived and those who were non-deprived and found both groups had equal increases in shuttle run distance [40]. One study found small increases in fitness at both four and 12 months in favour of the control group (p = 0.048), however there were high levels of missing data, and this result was not statistically significant when only complete cases were analysed or when imputed values were used [20].

**Table 4. Study outcome measures, measurement tools, main findings and general comments.**

| Authors (year of study) | Outcome measure (s) and measurement tool | Main findings | General comments |
|---|---|---|---|
| Arkesteyn et al. 2022 [30] | Self-perceived competence &self-esteem- (SPPC self-reported). Mental health (SDQ parents complete) | Small but significant increase in perceived global self-worth (p = .041)<br>Children with low baseline SPPC scores showed significant increases with large effect sizes for global self-worth (p = < .001), scholastic competence (p = .001), social competence (p = .003), athletic competence (p = .002), physical appearance (p = < .001) and behavioural conduct (p = .003)<br>Total difficulties score- no interaction effect for time x gender.<br>Significant reductions over time reported by parents for total difficulties (p < .001), hyperactivity (p = .004), peer problems (p = .008) and emotional symptoms (p = < .001) | Most increases occurred between week 10 and week 20. Compliance was monitored- 2 schools participated 3x/week 2 schools 4x/week 3 schools 5x/week |
| Booth et al. 2022 [31] | Cognition: Inhibition- (stop-signal task), visual spatial working memory-(static boxes search task), verbal working memory (reading span task) (self-completed on computer).<br>Subjective wellbeing- (Adapted Children's Feeling scale and Felt Arousal Scale children self-report). Fitness (20m shuttle run test child complete) | Significant difference in visual spatial working memory scores in unadjusted models. Longer term group significantly higher scores in visual working memory (adjusted for age, sex, SES) p<0.001, compared to those who did not participate in TDM<br>No statistically significant differences in wellbeing between those participating in TDM and those who did not take part<br>Longer Term participation group greater shuttle distance than the group who did not do the TDM p<0.05. And those who had shorter term participation, p<0.01. Remained statistically significant when adjusted for age, sex and SES | Longer term participation (more than 3 months) but was not possible to quantify further |
| Breheny et al. 2020 [20] | BMIz score at 12 months (British 1990 growth ref data) Fitness (British Athletics Linear Track Test)<br>Child reported QOL &Wellbeing (CHU9D & MDI Self-reported electronically under teacher supervision)<br>Academic Performance (teacher rated) | No significant impact on BMIz scores. Subgroup analysis showed significant interaction by sex- modest and statistically significant intervention effect on BMIz for girls at 12 months<br>Fitness: Small difference in favour of control group at both 4 and 12 months but not statistically significant for imputed or complete case analysis<br>QOL and Wellbeing: Small non-significant differences between groups in favour of intervention<br>Academic performance: Small difference in academic attainment in favour of intervention at 12 months (p = < 0.001). Only significant in complete case analysis and not after imputation | High levels of missing data in secondary outcomes |
| Brustio et al. 2019 [32] | Fitness (6-minute run test)<br>BMI | After correcting for age and gender significant group x time interactions were observed. TDM group showed an increased result between baseline and post-test (estimated difference = 25.15m, SE = 6.39m, p<0.001; percent change = 3.1%, compared with control group (estimated difference = 4.44m, SE = 6.69m, p = 0.911; percent change = 0.5%)<br>No significant group x time interactions were observed in BMI | On average, TDM was implemented 3x/week |
| Brustio et al. 2020 [33] | Fitness- (6-minute run test child complete)<br>Waist-to-height ratio<br>BMI | Fitness: Significant group*time interactions reported after correcting for age and BMI. I2 different T1-T2 and T1-T3. I3 different T2-T3 and T1-T3, but not T1-T2. Control different T2-T3 and T1-T3<br>Effect size greater for 3xweek (effect size 0.51) rather than 2xweek (effect size 0.29)<br>Waist-to-height ratio: Significant difference in group x time interaction effect, with I3 lower between pre and mid test.<br>No difference in BMI between groups | Data in results differs from abstract |

*(Continued)*

**Table 4.** (Continued)

| Authors (year of study) | Outcome measure (s) and measurement tool | Main findings | General comments |
|---|---|---|---|
| Chesham et al. 2018 [34] | MVPA (Accelerometer), Fitness (20m SRT, Body composition skin folds (Standard ISAK procedures) | MVPA relative increase of 9.1 minutes per day (95% CI 5.1min-13.2min) p = 0.027 Fitness: Total shuttle distance relative increase of 39.1m (95% CI 21.9 to 56.3m p = 0.037) Skin folds- relative decrease of 1.4mm (-2.0 to -0.8) p = 0.034 | Some methodological issues- Different duration and data collection points for control and intervention groups |
| Dring et al. 2022 [36] | Cognitive function (Stroop test, Sternberg paradigm, flanker task Children self-complete on laptop) Body composition (4 skinfold sites) Body mass and BMI (Age and sex-specific British 1990 growth reference). Physical fitness (MSFT-20m shuttle runs (child complete) | Stroop test- No difference in response times on the simple level. Response times on complex level intervention group significantly faster at follow up compared to control group p = 0.048. For accuracy no difference between intervention and control group at follow-up for either simple (p = 0.434) or complex (p = 0.580) levels Sternberg Paradigm and Flanker test- No difference for response times or accuracy at any level No difference between the intervention and control group in body composition, body mass or BMI Significant difference between the intervention and control group at follow-up for distance covered on MSFT. Intervention group 880m, compared to control group 740m p = 0.002 | |
| De Jonge et al. 2020 [35] | Fitness (SRT child complete) | Significant increase in SRT between control and intervention groups. The change in SRT score in the intervention group was significantly greater than the change intervention-plus group | Two intervention groups, Intervention plus group- additional support for teachers didn't make any difference |
| Harris et al. 2020 [37] | MVPA (SOFIT administered by one observer) | KS1 students- 100% of TDM in MVPA. Max time spent performing MVPA occurred when students interacted with peers & teachers promoted activity. KS2 students spend 13mins (88.1%) of TDM at MVPA | KS1 did TDM on 54/59 (91.5%) days. KS2 did TDM on 51/59 (86.4%) days |
| Hatch et al. 2021a [38] | Inhibitory control (Stroop test) Visual working memory (Sternberg paradigm) cognitive flexibility (Flanker test) All self-completed on laptop | No difference in response times between TDM and resting | |
| Hatch et al. 2021b [39] | Fitness- (Multi-stage fitness test child complete) | Highest fit children ran further than less fit children (main effect of fitness, p<0.001) | |
| Marchant et al. 2020 [40] | Fitness (20m SRT child complete) | Both groups equal increases in shuttle runs. No significant difference in shuttle run increase for deprived compared to non-deprived children when age and gender were adjusted for | Seasonal differences in data collection between schools |
| Morris et al. 2019 [19] | PA (Accelerometers) Maths fluency (MASSAT children complete) Executive function (Trail Making Task, Digit Recall, Flanker and Animal Stroop children self-complete 4 paper tests) | Children in TDM engaged in statistically significantly more MVPA p≤0.001. Achieving 10.67 ±2.74min of MVPA during TDM compared to the control (0.44±0.95min) Maths fluency: No significant improvements Executive function: No significant difference | |

I2- Intervention group 2x week Daily Mile participation. I3- Intervention group 3x week Daily Mile participation SPPC -Self-perception Profile for Children SDQ-The Strengths and Difficulties Questionnaire BMI- Body Mass Index SRT-Shuttle run test QOL- Quality of Life CHU9D-Child Health Utility 9 Dimension MYDI-Middle Years Development Instrument MVPA-Moderate to vigorous physical activity SOFIT-System for observing fitness instruction time ISAK- The International Society for the Advancement of Kinanthropometry. MSFT-Multistage fitness test PA- Physical activity MASSAT-Maths Addition and Subtraction, Speed and Accuracy Test

**Effects on physical health.** Three studies reported BMI pre and post intervention [32, 33, 36], however none of the studies found a significant difference in BMI between groups. In one study where BMIz scores (Body mass index z-scores, are a measure of relative weight adjusted for child age and sex) were reported [20] although TDM did not have a significant impact on BMIz scores at 12 months, subgroup analysis indicated significant interaction by sex, with the intervention effect for girls being modest and statistically significant at 12 months [20].

Three studies reported on body composition [33, 34, 36]. One study reported no difference in waist circumference, hip circumference, or sum of skinfolds, between intervention and control groups [36]. One study reported a relative decrease of 1.4mm in skin folds [34]. In one study, lower waist-to-height ratios were found between pre and mid test in the group who completed TDM more than 2.5 times a week on average [33].

**Effects on psychological wellbeing.** Psychological wellbeing was reported in two studies [20, 31], one study found small differences in favour of the intervention group after 12 months of TDM, but the results were not statistically significant [20] and the other study found no statistically significant differences in wellbeing between those participating in TDM and those who did not take part [31].

Small but significant increases in perceived global self-worth were found using The Self-Perception Profile for Children (SPPC) [30]. Children with low baseline SPPC scores showed significant increases with large effect sizes for global self-worth, scholastic competence, social competence, athletic competence, physical appearance and behavioural conduct [30].

**Effects on mental health.** One study [30] reported the impact of participation in TDM on mental health, as measured by The Strengths and Difficulties Questionnaire [41] completed by their parents. There were significant reductions over time reported by parents for total difficulties, hyperactivity, peer problems and emotional symptoms [30].

**Effects on academic performance.** Two studies reported the effect of TDM on academic performance [19, 20]. One study found a small difference in teacher rated academic attainment in favour of the intervention group at 12 months, however there was high levels of missing data (over 50%), and this was only significant when complete cases were analysed and not when imputed values were used [20]. One study found no significant improvements in maths fluency scores after a single bout of TDM [19].

**Effects on cognition.** Two studies included explored the effects of an acute bout of TDM on various aspects of cognitive function [19, 38]. No significant improvements were found in executive function [19], inhibitory control [38], cognitive flexibility [38] or working memory [38] after a single bout of TDM.

However, one study where children had longer term participation in TDM (3 months or more) found that those who had participated in TDM for longer had higher scores in visual spatial working memory than those who did not participate in TDM [31]. In addition, although one study did not find five weeks participation in TDM to increase response times in the simple level in the Stroop test, response times were significantly faster on more complex versions of the Stroop test in the intervention group [36].

## Discussion

This systematic review summarised the results from 13 studies, examining the impact of TDM on children's physical activity levels, physical fitness, physical health, psychological wellbeing, academic performance, and cognitive function. To the best of our knowledge, this is the first systematic review of TDM initiative for primary school-aged children. Over the past ten years there has been a rapid adoption of TDM and other 'active miles' in schools, community settings, and such approaches have been cited in, and formed the basis of Government policy.

This review was conducted in response to the limited evidence base to support such widescale adoption, and to guide the future integration of TDM into health interventions and potential policy. All studies included in this review were also assessed for methodological quality to examine what degree of confidence could be placed alongside study outcomes.

Overall, both acute and longer-term participation in TDM was found to increase MVPA by approximately ten minutes per-day [19, 34]. This is a relatively greater increase in MVPA than found by other PA interventions in primary school children [42, 43]. Although findings show that children do not spend the whole 15-minutes of TDM at a moderate-to-vigorous intensity [19, 34, 37], the reported increases are welcomed given higher levels of MVPA have been associated with improved cardiometabolic health in children [44]. Additionally, it goes some way to help children achieve the public health recommended 60 minutes of MVPA per day.

In addition to an increase in MVPA, in general, the included studies reported a positive effect of TDM on physical fitness, however, the variety of different fitness tests used across studies makes direct comparisons between studies difficult. One of the studies which lasted for 5 weeks, reported the intervention group completed 140 metres more than the control group in the multi-stage fitness test at follow-up [36]. This shows that improvements in fitness can be achieved in a relatively short period of time when TDM is conducted five days per-week [36]. Frequency of participation is an important factor to consider, with those who participated at least twice a week showing an increase of 5.6% in a 6-minute run test, whereas those who performed TDM three or more times a week had an increase of 8.8% [33]. These results suggest there may be a dose-response associated with TDM, requiring implementation according to the core principles (performed least 3x/week) to maximise improvements in physical fitness.

None of the included studies reported a significant reduction in BMI. This is in contrast with a Cochrane Review which found that PA interventions can reduce BMI in children aged six to 12 years old [45]. This likely suggests the effect of 15 minutes of daily (3–4 times week) exercise is not enough to substantially impact weight. However, as children's body composition is naturally changing at this developmental stage, a reduction in BMI should not be a primary aim of TDM implementation.

Longer-term (20 weeks) participation in TDM was reported to improve children's mental health as measured by The Strengths and Difficulties Questionnaire, along with small but significant increases in perceived global self-worth [30]. The greatest increase occurred between weeks 10 and 20, suggesting that the changes in self-esteem may only take place when the PA in school is sustained for a longer period of time [30]. In addition, the finding that children with lower baseline SPPC scores had large positive increases in perceptions of competence, physical appearance and behavioural conduct (see Table 4) indicates that the effect of TDM may be greater for children with lower initial perceptions of self-worth and self-competence. Although these results are promising, especially with schools adopting TDM into their COVID-19 recovery plans for children's mental health and wellbeing, the results were found in a single arm pilot study so there was not a control group for comparison.

More research is needed with regards to the effect of TDM on academic performance, with only two studies reporting outcomes of academic performance [19, 20]. In one study [20] academic performance was measured through teacher reported scores, which has potential for bias. In addition, there was a large degree of missing data (over 56%) and therefore these results should be interpreted with caution.

This review found that a single bout of TDM did not have a significant effect on cognition. This is in contrast with two systematic reviews which found the majority of acute PA interventions in children improved cognitive function [46, 47]. These results suggests that 15 minutes of TDM running exercise may not be enough to impact cognition and a longer bout of exercise may be required to see benefits, or there may have been methodological issues that did not

capture the potential effects. However, longer term participation in TDM improved visual spatial working memory [31] and increased response times on complex levels in the Stroop test [36]. However, the latter finding was based on a small study (n = 79) with a quasi-experimental design. Consequently, larger, randomised controlled trials are needed to strengthen the evidence base of what, if any, effects are present for TDM on cognition.

Many of the findings suggest that timing of data sampling and the length of intervention are important factors to consider. In one study, improvements in waist-to-height ratios in the intervention group were reported at three months but at six months they had reverted to similar scores to baseline [33]. No formal process evaluation was carried out so it is not possible to assess how often TDM was performed and how this may have changed over the duration of the study. These results may suggest that compliance and motivation dropped off after the initial excitement of participation or that TDM only has benefits initially [20]. Given the implications for both research recommendations and TDM implementation in practice, future studies should record compliance and report on intervention fidelity throughout the intervention period.

One study reported that although children enjoyed participating in TDM they expressed an appeal for more variety in activity types and described TDM as 'a bit boring' [38]. This was also reported by Marchant and colleagues [40] where pupils discussed one of the barriers to TDM being lack of enjoyment and boredom associated with it, suggesting that after initial excitement wore off motivation decreased [40].

One of the key attractive features of TDM is the simplicity of adoption and delivery by teachers, with no equipment or special training required. However, research into the barriers and facilitators of TDM have found that approximately half the teachers use some form of reward system to increase motivation [48]. Different methods include awarding tokens, tracking distance, or teachers running with pupils [48]. According to theories of motivation (i.e., Self-Determination Theory [49]), if this is done in an outcome-orientated, controlling style, it may undermine longer term autonomous motivation for PA. Additional planning and preparation by teachers can add to their already heavy workload and may result in waning participation in TDM. Qualitative research supports this interpretation, suggesting TDM may not be as simple to implement in practice with additional costs associated with extra staff time to prepare for an engaging and exciting Daily Mile experience [50]. Teachers reported the need to keep TDM 'fresh' by adding new motivational strategies to keep pupils engaged [50].

The intensity TDM is performed at is another important consideration. The majority of longer-term studies failed to measure or report the intensity that TDM was performed. As a result, it is not possible to report the effect of intensity on the outcomes in most studies. It is probable that the studies in which the children performed TDM at a higher intensity saw greater improvements in physical fitness. One study reported a large variation in intensity with the most active children spending the duration of TDM in MVPA, compared to the least active children who only spent 33% at MVPA [19]. More attention in future research could assess intensity, and if intensity levels are implicated in affective experiences of TDM [51].

There is also a concern that the long-term sustainability of TDM is limited due to the lack of behaviour change theory principles underpinning it in the school environment. It has been suggested that it may be beneficial to develop a programme theory in order to help understand and explain the behaviour of staff and pupils involved in TDM [52]. Physical activity interventions which were theoretically underpinned have been found to have the greatest effect on long-term behaviour change [53]. More so, the potential theoretical underpinnings of TDM should consider principles from a broad range of approaches, including social-cognitive, humanistic, dual-process, and socioecological frameworks [54].

Another factor which needs to be considered in further research is the need to include children with physical or intellectual disabilities. These children were not included in many of the studies in this review, or were excluded from analysis, despite one of TDM core principles being inclusivity. Children with intellectual or physical disabilities are an important group to target as are often reported as being less fit and have poorer health than their non-disabled peers [55]. The 'Walk-Buds' trial which is currently underway, operates a peer buddy system, where younger children are partnered with an older peer with similar interests to complete their physical activity [56]. A similar approach may be worth considering with TDM. Furthermore, evaluation into the impact of TDM for children with intellectual or physical disabilities is needed to provide evidence of the benefits for these groups.

## Strengths and limitations

This systematic review followed the PRISMA guidelines, and all papers were screened independently by two authors. Data extraction was done by one author and checked by a second, and quality assessment was completed independently by two reviewers. This review implemented a comprehensive search strategy which was developed by the authors and institution librarian. A further strength of the study was that it looked at both the acute and long-term effect of TDM. However, there were limitations. First, this systematic review included all study designs: randomised controlled trials, quasi-experimental studies, pilot studies, repeat measures, cross-sectional, and natural experiments. We considered it important to keep the inclusion criteria broad to explore the scope and quality of the existing evidence-base supporting the benefits of TDM and the overall inclusion of only 13 studies in the review justifies this decision. However, it also highlights a dearth of high-quality design investigations (e.g., RCTs) of TDM (see Table 3) that, in turn, limit the strengths of our conclusions. As such, the evidence-base and conclusions presented in the review should be considered in that light. There was also a high level of heterogeneity between the studies included in the review and therefore it was difficult to compare studies directly. One such difference was how TDM was implemented; for example, the duration of TDM, how many times a week it was performed and the intensity at which it was performed. Moreover, there were large variations in both the reporting of intervention fidelity and the compliance to the intervention. Future studies need to record and report fidelity and compliance data at an individual level as it may be a potential confounder.

Most studies included in this review involved schools who had self-selected themselves to take part, it is likely that schools with staff who have an interest in PA and are aware of the benefits and importance of it were more likely to take part and were more motivated to facilitate TDM and as a result there could be some sampling bias.

## Conclusion and future directions

The Daily Mile is performed in over 15,600 schools and nurseries across the world [17], in addition to underpinning some public health policy. However, the evidence supporting its benefits is limited to a relatively small number of mostly fair-to-good quality studies (n = 13). As such, this systematic review has gone some way to clarify the quality and robustness of this existing evidence base. Findings from this review suggest TDM can increase children's physical fitness and MVPA levels. There is also some fair-to-good quality evidence that it may improve body composition, mental health, and self-perceptions, however null effects were observed for BMI and academic performance. An acute bout of TDM did not affect cognitive function, although longer term participation was found to improve some areas of cognitive function. Whilst initial results are promising, the long-term benefits are unclear due to an insufficient

number of studies reported, and a dearth of good and excellent quality research study designs across each of the outcomes reported. Future research should also explore other indicators of health status in children not considered to date, such as bone health or muscle strength, for example. As such, higher quality research is needed with longer term follow up to explore the sustainability of intervention effects. There is also a need for process evaluations and proper reporting to ascertain implementation.

Promisingly, a longitudinal quasi-experimental cohort study is currently underway, following children from year 1 (age 5/6) to year 6 (age 10/11) [57]. It is hoped this, and future likewise studies will provide greater evidence of the long-term benefits of TDM not only on PA levels, but on the other outcomes covered in this review. In addition, as only four of the included studies employed any randomisation, future randomised controlled trials are required to ensure that any public policy recommendations for TDM on improving mental health, wellbeing, cognitive function and academic achievement are centred on a stronger scientific evidence base.

## Supporting information

**S1 Checklist. PRISMA 2020 checklist.**
(DOCX)

**S1 File. Screening tool for independent author screening.**
(DOCX)

**S2 File. Search from PsycINFO.**
(DOCX)

**S3 File. Template data collection form.**
(DOCX)

## Acknowledgments

We would like to acknowledge the support of the Daily Mile Network Northern Ireland, Nicola Topping from the Education Authority Northern Ireland and Colette Brolly from the Public Health Agency in Northern Ireland.

## Author Contributions

**Conceptualization:** Gavin Breslin.

**Data curation:** Medbh Hillyard.

**Formal analysis:** Medbh Hillyard.

**Funding acquisition:** Gavin Breslin.

**Investigation:** Gavin Breslin, Medbh Hillyard, Noel Brick, Stephen Shannon, Brenda McKay-Redmond, Barbara McConnell.

**Methodology:** Gavin Breslin, Medbh Hillyard.

**Writing – original draft:** Medbh Hillyard.

**Writing – review & editing:** Gavin Breslin, Medbh Hillyard, Noel Brick, Stephen Shannon, Brenda McKay-Redmond, Barbara McConnell.

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
