## [Decision Letter · Decision Letter 0]

12 Dec 2022

PONE-D-22-29361A rapid systematic review of the effect of The Daily Mile™ on children's physical activity, physical health, mental health, wellbeing, academic performance and cognitive functionPLOS ONE

Dear Dr. Medbh Hillyard,

Thank you for submitting your manuscript to PLOS ONE. After careful consideration, we feel that it has merit but does not fully meet PLOS ONE’s publication criteria as it currently stands. Therefore, we invite you to submit a revised version of the manuscript that addresses the points raised during the review process.

Please submit your revised manuscript by December 29, 2022. If you will need more time than this to complete your revisions, please reply to this message or contact the journal office at plosone@plos.org. Please include the following items when submitting your revised manuscript:A rebuttal letter that responds to each point raised by the academic editor and reviewer(s). You should upload this letter as a separate file labeled 'Response to Reviewers'.A marked-up copy of your manuscript that highlights changes made to the original version. You should upload this as a separate file labeled 'Revised Manuscript with Track Changes'.An unmarked version of your revised paper without tracked changes. You should upload this as a separate file labeled 'Manuscript'.If applicable, we recommend that you deposit your laboratory protocols in protocols.io to enhance the reproducibility of your results. Protocols.io assigns your protocol its own identifier (DOI) so that it can be cited independently in the future. For instructions see: https://journals.plos.org/plosone/s/submission-guidelines#loc-laboratory-protocols. Additionally, PLOS ONE offers an option for publishing peer-reviewed Lab Protocol articles, which describe protocols hosted on protocols.io. Read more information on sharing protocols at https://plos.org/protocols?utm_medium=editorial-email&utm_source=authorletters&utm_campaign=protocols.

We look forward to receiving your revised manuscript.

Kind regards,

Cosme F. Buzzachera, Ph.D.

Academic Editor

PLOS ONE

Journal Requirements:

"I have read the journal's policy and the authors of this manuscript have the following competing interests: All authors are members of The Daily Mile Network Northern Ireland. "

Additional Editor Comments:

The authors are commended for an interesting manuscript. Minor modifications are needed to be made. All modifications can be made at this moment. Please see my comments below.

In the Background section, a detailed explanation of the TDM principles is strongly required. An example can be found here:

"Hanckel B et al. The Daily Mile as a public health intervention....BMC Public Health, 2019."

The keywords used for the systematic review can be found in the supplementary files. However, I suggest including it in the Methods section, indicating the presence of the full strategy explanation in the supplementary file.

The selection of studies (for example, Morris et al. 2019) exploring the acute effects of TDM is important. How can this decision bias the results of more robust studies using pre- and post-interventions? A brief comment is necessary for the Limitations section.

Last, what does a "rapid" systematic review mean? Is the term "rapid" necessary in the Title?

Reviewers' comments:

Reviewer's Responses to Questions

**Comments to the Author**

1. Is the manuscript technically sound, and do the data support the conclusions?

Reviewer #1: Yes

Reviewer #2: Yes

2. Has the statistical analysis been performed appropriately and rigorously? 

Reviewer #1: N/A

Reviewer #2: Yes

3. Have the authors made all data underlying the findings in their manuscript fully available?

Reviewer #1: Yes

Reviewer #2: Yes

4. Is the manuscript presented in an intelligible fashion and written in standard English?

Reviewer #1: Yes

Reviewer #2: Yes

5. Review Comments to the Author

Reviewer #1: Title: PONE-D-22-29361 A rapid systematic review of the effect of The Daily Mile™ on children's physical activity, physical health, mental health, wellbeing, academic performance and cognitive function

Summary: The study under review investigated the effect of the Daily Mile™ (TDM) on children’s physical activity levels, physical health, mental health, wellbeing, academic performance, and cognitive function. Six databases were systematically searched, and 13 eligible studies involving school-aged children (aged 4-12 yrs) taking part in TDM were inserted. Longer-term participation in TDM was associated with increased MVPA and improved physical fitness. However, no significant changes in BMI, academic performance, and wellbeing were reported, with limited effects on cognitive function and mental health. The authors concluded that TDM increases physical activity and physical fitness. The authors noted, however, that "long-term improvement on mental health, wellbeing, academic performance, and cognitive function requires further good-to-excellent quality research."

General comments: The arguments for the manuscript under review are original and exciting. There are minor concerns and methodological issues with the manuscript in its current form. For your information, I have attached my comments below. I hope you will find them to be constructive and helpful.

Minor Concerns: The primary concerns with the manuscript are presented below in order of appearance:

Background.

The background is well-written, except for this sentence - lines 112-114. Please revise. The study's problem, hypothesis, and purposes are clear to the reader.

Methods.

Line 144. The eligibility of the studies should be clarified more. Were utilized only randomized or pseudo-randomized controlled trial (RCT) designs with pre- and post-assessment? Were observational studies, review articles, published abstracts, and case studies included or excluded? Please clarify. Also, explain why control or comparison groups were not required. This decision is a limitation and should be recognized.

Line 166 (Outcomes). Key definitions should be clarified for consistency (for details, see Watson et al. 2017, pg. 3; https://doi.org/10.1186/s12966-017-0569-9). Such definitions are cognitive function, academic achievement, and wellbeing.

Results.

Line 258. What is the importance of considering studies that analyzed only "one session of TDM"? Please comment, and if necessary, recognize it as a limitation.

Miscellaneous.

Line 75. Please replace 'per-day' with 'per day.'

Line 76. MVPA abbreviation should be included here.

Line 78. Insert a comma after "[7]".

Line 85. Insert a point after "[12]".

Line 89. Please replace 'decreases' with 'decrease.'

Line 89. Please replace 'involves' with 'involved.'

Line 106. Please insert the TDM website.

Line 112. Please replace 'short term' with 'short-term.'

Line 112. Please replace 'follow up' with 'follow-up.'

Line 169. Please replace 'bleep' with 'beep.'

Line 170. Physical health is not synonymous with weight, BMI, and body composition. Such terms are more related to nutritional status/health.

Line 265. What is "This" referred to? Please revise.

Line 316. Remove the comma after 'group' and insert 'and' before 'the difference.'

Line 317. Insert 'and' before 'for MVPA.'

Line 318. What do 'Key Stage 1' and 'Key Stage 2' mean?

Line 324. Please replace 'use;' with "used" and insert "and included" or "as such."

Line 325. Aren't the "20m shuttle run test" and the "Multi-stage fitness test" the same test?

Line 344. Please replace "skin folds" with "skinfolds."

Reviewer #2: It is a review article with a very relevant subject, which followed the rules for preparing a systematic review. I suggest modify the keywords that are already in the title by others equally indexed, but different (i.e., daily mile, physical activity).

6. PLOS authors have the option to publish the peer review history of their article (what does this mean?). If published, this will include your full peer review and any attached files.

Reviewer #1: No

Reviewer #2: **Yes: **FABIANA ANDRADE MACHADO

---

## [Author Response · Author response to Decision Letter 0]

21 Dec 2022

Thank for you providing this information, we have made any changes required based on PLOS ONE’s style requirements. 

"I have read the journal's policy and the authors of this manuscript have the following competing interests: All authors are members of The Daily Mile Network Northern Ireland. "

We have included the updated competing interests statement in our cover letter.

Reference list has been checked to ensure it is complete and correct. 

Additional Editor Comments:

The authors are commended for an interesting manuscript. Minor modifications are needed to be made. All modifications can be made at this moment. Please see my comments below.

Thank you for the kind and very helpful comments, we have made the requested modifications.

In the Background section, a detailed explanation of the TDM principles is strongly required. An example can be found here:

"Hanckel B et al. The Daily Mile as a public health intervention....BMC Public Health, 2019."

A detailed description of the core principles have been included in lines 110-116.

The keywords used for the systematic review can be found in the supplementary files. However, I suggest including it in the Methods section, indicating the presence of the full strategy explanation in the supplementary file.

As requested the search terms have been included in the methods section in Table 1 (p9). 

The selection of studies (for example, Morris et al. 2019) exploring the acute effects of TDM is important. How can this decision bias the results of more robust studies using pre- and post-interventions? A brief comment is necessary for the Limitations section.

We thank the reviewer for this comment. We do not consider that the inclusion of acute studies (e.g., Morris et al., 2019) adds a bias to the results. Throughout the results section we are explicit about the distinction between acute (e.g., P25, L439-442) and longer term (e.g., P25, L444-447) studies and, where relevant, the comparable effects of both acute and longer-term studies (e.g., p. 26, L427). We also highlight disparity between the findings of our review and previous reviews for acute bouts of activity (e.g., P28, L743-748), and also how the findings of acute and longer-term studies included within this review differ in terms of their findings (e.g., P28, L748-449). As such, while we would agree that the inclusion of both acute and longer-term studies would be problematic if we did not separate the outcomes of each study design within our results and discussion. However, because we have done so, we consider it a strength of this review that adds to what we currently know about the acute and longer-term effects of TDM on the selected outcomes. This has been expressed within the Strengths and Limitations section (P31, L837-838).

Last, what does a "rapid" systematic review mean? Is the term "rapid" necessary in the Title?

We agree with the comment and rapid has been removed from the title. 

Reviewers' comments:

Reviewer's Responses to Questions

Comments to the Author

1. Is the manuscript technically sound, and do the data support the conclusions?

Reviewer #1: Yes

Reviewer #2: Yes

2. Has the statistical analysis been performed appropriately and rigorously? 

Reviewer #1: N/A

Reviewer #2: Yes

3. Have the authors made all data underlying the findings in their manuscript fully available?

Reviewer #1: Yes

Reviewer #2: Yes

4. Is the manuscript presented in an intelligible fashion and written in standard English?

Reviewer #1: Yes

Reviewer #2: Yes

5. Review Comments to the Author

Reviewer #1: Title: PONE-D-22-29361 A rapid systematic review of the effect of The Daily Mile™ on children's physical activity, physical health, mental health, wellbeing, academic performance and cognitive function

Summary: The study under review investigated the effect of the Daily Mile™ (TDM) on children’s physical activity levels, physical health, mental health, wellbeing, academic performance, and cognitive function. Six databases were systematically searched, and 13 eligible studies involving school-aged children (aged 4-12 yrs) taking part in TDM were inserted. Longer-term participation in TDM was associated with increased MVPA and improved physical fitness. However, no significant changes in BMI, academic performance, and wellbeing were reported, with limited effects on cognitive function and mental health. The authors concluded that TDM increases physical activity and physical fitness. The authors noted, however, that "long-term improvement on mental health, wellbeing, academic performance, and cognitive function requires further good-to-excellent quality research."

General comments: The arguments for the manuscript under review are original and exciting. There are minor concerns and methodological issues with the manuscript in its current form. For your information, I have attached my comments below. I hope you will find them to be constructive and helpful.

Thank you for the constructive and positive comments. 

Minor Concerns: The primary concerns with the manuscript are presented below in order of appearance:

Background.

The background is well-written, except for this sentence - lines 112-114. Please revise. The study's problem, hypothesis, and purposes are clear to the reader.

The sentence has been revised to make it clearer (See P6, L133-135).

Methods.

Line 144. The eligibility of the studies should be clarified more. Were utilized only randomized or pseudo-randomized controlled trial (RCT) designs with pre- and post-assessment? Were observational studies, review articles, published abstracts, and case studies included or excluded? Please clarify. Also, explain why control or comparison groups were not required. This decision is a limitation and should be recognized.

This has been clarified in lines 188-189. We have also added to the limitations (P31, L838-8846) to highlight this limitation. 

Line 166 (Outcomes). Key definitions should be clarified for consistency (for details, see Watson et al. 2017, pg. 3; https://doi.org/10.1186/s12966-017-0569-9). Such definitions are cognitive function, academic achievement, and wellbeing.

Definitions have been included in lines 203-213. 

Results.

Line 258. What is the importance of considering studies that analyzed only "one session of TDM"? Please comment, and if necessary, recognize it as a limitation.

Thank you for this comment. It is important to consider both acute and long-term effects of any exercise intervention, even if that is one bout of activity. Previous meta-analytic reviews, for example, have highlighted the acute effects of activity: See 

Reed, J., & Ones, D. S. (2006). The effect of acute aerobic exercise on positive activated affect: A meta-analysis. Psychology of Sport and Exercise, 7(5), 477–514.

and the long-term effects on psychological wellbeing, see:

Reed, J., & Buck, S. (2009). The effect of regular aerobic exercise on positive-activated affect: A meta-analysis. Psychology of Sport and Exercise, 10(6), 581–594.

In line with previous research, we wanted to ensure we showed the acute and long-term effects of the TDM. Specific to TDM these include both acute (e.g., does a single bout of TDM increase daily MVPA; does an acute bout impact on post-activity cognitive function) and chronic (e.g., does longer-term TDM participation increase physical fitness; does TDM impact on longer-term physical and mental health). As such, our data reporting in our view and in line with previous research is not considered a limitation, but a strength of our review. We also commented on the following in our responses to the Editor:

Do not consider that the inclusion of acute studies adds a bias to the results. Throughout the results section we are explicit about the distinction between acute (e.g., P25, L439-442)) and longer term (e.g., P25, L444-447) studies and, where relevant, the comparable effects of both acute and longer-term studies (e.g., p. 26, L427). We also highlight disparity between the findings of our review and previous reviews for acute bouts (e.g., P28, L743-748), and also how the findings of acute and longer-term studies included within this review differ in terms of their findings (e.g., P28, L472-481).

As such, while we would agree that the inclusion of both acute and longer-term studies would be problematic if we did not separate the outcomes of each design within our results and discussion. However, because we have done so, we consider it a strength of this review that adds to what we currently know about the acute and longer-term effects of TDM on the selected outcomes. This has been expressed within the Strengths and Limitations section (P31, L837-838).

Miscellaneous.

Line 75. Please replace 'per-day' with 'per day.'

Done 

Line 76. MVPA abbreviation should be included here.

Done

Line 78. Insert a comma after "[7]". 

Done

Line 85. Insert a point after "[12]".

Done

Line 89. Please replace 'decreases' with 'decrease.'

Done

Line 89. Please replace 'involves' with 'involved.'

Done

Line 106. Please insert the TDM website.

Done

Line 112. Please replace 'short term' with 'short-term.'

Done

Line 112. Please replace 'follow up' with 'follow-up.'

Done

Line 169. Please replace 'bleep' with 'beep.'

Both “bleep” and “beep” are commonly used to refer to this test. However, to avoid confusion, we have amended this to the “Multistage Fitness Test” to give this test it’s correct title.

Line 170. Physical health is not synonymous with weight, BMI, and body composition. Such terms are more related to nutritional status/health.

Thank you for this insightful comment. We agree with the reviewer that weight, BMI and Body composition are not synonymous with physical health but can be used as markers of physical health. We also consider that other markers of physical health, in addition to indicators of overweight and obesity, should be considered in the research. These might include bone health (e.g., bone mass, bone mineral density), or muscle strength, for example. However, to date, the only indicators of physical health status measured in this area have been measures of overweight and obesity. To clarify this, we have added “indicators of physical health status” to the Outcomes section (P8, L200). And added a further note to the future research directions section (L874-876) to read, “Future research should also explore other indicators of health status in children not considered to date, such as bone health or muscular strength, for example.”

Line 265. What is "This" referred to? Please revise.

Updated and wording has been changed here in light of your comment. 

Line 316. Remove the comma after 'group' and insert 'and' before 'the difference.'

Changes made

Line 317. Insert 'and' before 'for MVPA.'

Done

Line 318. What do 'Key Stage 1' and 'Key Stage 2' mean?

Ages of Key stage 1 and 2 included. 

Line 324. Please replace 'use;' with "used" and insert "and included" or "as such."

Done

Line 325. Aren't the "20m shuttle run test" and the "Multi-stage fitness test" the same test?

Changed to reflect this. 

Line 344. Please replace "skin folds" with "skinfolds."

Done

Reviewer #2: It is a review article with a very relevant subject, which followed the rules for preparing a systematic review. I suggest modify the keywords that are already in the title by others equally indexed, but different (i.e., daily mile, physical activity).

Key words have been changed so any mentioned in the title are not repeated as key words.

6. PLOS authors have the option to publish the peer review history of their article (what does this mean?). If published, this will include your full peer review and any attached files.

Do you want your identity to be public for this peer review? For information about this choice, including consent withdrawal, please see our Privacy Policy.

Reviewer #1: No

Reviewer #2: Yes: FABIANA ANDRADE MACHADO

---

## [Editor Report · Decision Letter 1]

26 Dec 2022

A systematic review of the effect of The Daily Mile™ on children's physical activity, physical health, mental health, wellbeing, academic performance and cognitive function.

PONE-D-22-29361R1

Dear Dr. Medbh Hillyard,

We’re pleased to inform you that your manuscript has been judged scientifically suitable for publication and will be formally accepted for publication once it meets all outstanding technical requirements.

Kind regards,

Cosme F. Buzzachera, Ph.D.

Academic Editor

PLOS ONE

Additional Editor Comments:

Dear Author.

Thank you for addressing all of the reviewers' comments.

The current version of the manuscript is, in my view, ready for publication.

Congratulations.

---

## [Editor Report · Acceptance letter]

3 Jan 2023

PONE-D-22-29361R1 

A systematic review of the effect of The Daily Mile™ on children's physical activity, physical health, mental health, wellbeing, academic performance and cognitive function 

Dear Dr. Hillyard:

I'm pleased to inform you that your manuscript has been deemed suitable for publication in PLOS ONE. Congratulations! Your manuscript is now with our production department. 

Kind regards, 

on behalf of

Dr. Cosme F. Buzzachera 

Academic Editor

PLOS ONE